# Relative Water Content, Chlorophyll Index, and Photosynthetic Pigments on *Lotus corniculatus* L. in Response to Water Deficit

**DOI:** 10.3390/plants13070961

**Published:** 2024-03-26

**Authors:** Luis Ángel González-Espíndola, Aurelio Pedroza-Sandoval, Ricardo Trejo-Calzada, María del Rosario Jacobo-Salcedo, Gabino García de los Santos, Jesús Josafath Quezada-Rivera

**Affiliations:** 1Universidad Autónoma Chapingo, Unidad Regional Universitaria de Zonas Áridas, Km 40 Carretera Gómez Palacio—Chihuahua, Bermejillo C.P. 35230, Durango, Mexico; qfbgonzalez_espindola@hotmail.com (L.Á.G.-E.); rtrejo@chapingo.uruza.edu.mx (R.T.-C.); jacobo.rosario@inifap.gob.mx (M.d.R.J.-S.); 2Colegio de Postgraduados, Campus Montecillo, Km 36.5 Carretera México-Texcoco, Montecillo C.P. 56230, Texcoco, Mexico; garciag@colpos.mx; 3Facultad de Ciencias Químicas, Universidad Juárez del Estado de Durango, Av. Universidad s/n. Fraccionamiento Filadelfia, Gómez Palacio, Durango C.P. 35010, Mexico; josafath.quezada@ujed.mx

**Keywords:** plant physiology, drought, plant water stress, plant performance, forage

## Abstract

This study aimed to evaluate different *L. corniculatus* L. ecotypes under water-deficit conditions to identify changes in relative water content and photosynthetic pigments as indicators of physiological responses during different years’ seasons. The experiment was conducted in a randomized block design with three replicates. Ten treatments were performed as a factorial of 2 × 5, where the first variation factor was the soil water content—no water deficit (NDW) with 100% field capacity (FC), and water deficit (DW) corresponding to 85.4% of the FC—and the second variation factor comprised four ecotypes and one variety of *L. corniculatus*. A significant effect was identified on the concentration of photosynthetic pigments, mainly total chlorophyll, with chlorophyll a in the 255301 ecotype with records of 187.8, 167.5, and 194.6 mg g^−1^ FW in WD, corresponding to an increase of 86.0%, 172.6%, and 16.6%, respectively, in relation the lower values obtained in the ecotype 202700 under NWD. In carotenoids, higher concentrations were observed in the 255301 and 202700 ecotypes and the Estanzuela Ganador variety under WD in most seasonal periods, except summer; a similar response was found in the 202700 ecotype and the Estanzuela Ganador variety during the winter season, also in WD. The results showed that the first two principal components accounted for 71.8% of the total variation, with PC1 representing chlorophyll *a*, chlorophyll *b*, and total chlorophyll, and PC2 representing carotenoids, temperature, relative chlorophyll index, and relative water content. The observations were grouped based on soil moisture content, with the optimal moisture group exhibiting higher chlorophyll and carotenoid concentrations. The findings suggest that soil moisture content significantly affects the performance of *L. corniculatus* ecotypes, and the plant shows seasonal variations in response to water-deficit conditions. This research contributes to understanding the physiological responses of *L. corniculatus* and its potential as a water-efficient forage crop for promoting sustainable agriculture and enhancing food security.

## 1. Introduction

Agriculture worldwide depends undoubtedly on the availability of water, which is scarce, putting food security at risk [1]. Additionally, the increase in population demands significant quantities of food, meat, and milk, which come from livestock farming [2]. This means high demand for forages, grasses, and legumes, which are essential food sources for livestock activity [3]. Also, this production system demands large volumes of water in irrigation areas, which are increasingly affected by the greater intensity of droughts due to climate change [4]. Furthermore, global climate change shows its impact through extreme temperature and precipitation events that adversely affect agri-food production in different regions of the planet [5,6]. Arid areas have recorded temperatures above or below the historical average during the summer and winter seasons. Droughts have worsened in time and space, affecting rainfed and irrigated agriculture and livestock [7,8].

Irrigated agricultural areas of northern Mexico are the leading suppliers of forage for stabled livestock—mainly alfalfa, maize, and sorghum forage. The production of these crops is affected by the low availability of water, the aquifer’s low recharge, and the water resource’s overexploitation. Notably, alfalfa is widely recognized for its high nutritional value as a forage food; however, it has a high rate of water demand, estimated at an average of 2 m of annual irrigation depth or greater [4,9]. The production of crops with high water demand in regions with aquifer depletion and intensive agricultural production is the main challenge to be overcome through technology that allows for mitigating social and environmental impacts [10,11].

Several studies have been concentrated on developing and analyzing diverse crop species that exhibit tolerance to various abiotic stressors, notably drought, salinity, light intensity, extreme temperatures, and heavy metals, either individually or in combination [12]. These intensive agricultural production systems with low availability and poor-quality water are being contaminated with salts and heavy metals (arsenic, cadmium, and lead) due to being extracted from great depths of the subsoil [13]. Therefore, the importance of options that reduce the environmental impact caused by the intensive use of water–soil–plant resources is essential in managing agri-food production systems sustainably. From this perspective, alternative crops with lower water demand and adequate productive competitiveness with other crops, such as alfalfa, are viable alternatives for agricultural areas with low availability of water resources [14]. Hence, exploring native or exotic crops with the potential to adapt to different environments is required. Generally, research is focused on studying tolerance to water stress, extreme temperatures, and biodiversity in agroecosystems [15].

In recent research, the legume *Lotus corniculatus* L., commonly known as birdsfoot trefoil, is being studied as an alternative forage crop [16]. This plant is a perennial native to Europe and has spread to different parts of the world due to its adaptative capability to different environments and desirable characteristics as a forage [17,18,19]. *L. corniculatus* is a leguminous plant that can fix nitrogen, thus improving soil fertility [20], and its large root system allows it to access water from deeper into the soil with the efficient use of hydric resources [21]. In addition, *L. corniculatus* has high nutritional value since its foliage is abundant in protein and is very tasty for livestock. Although this species of forage crop comes from temperate–humid environments, its great genetic diversity enables it to grow and develop adequately in conditions of extreme environments, such as arid zones [22]. Drought stress is the primary abiotic factor constraining plant growth, distribution, and survival, significantly limiting forest productivity in different ways such as the inhibition of photosynthesis and osmoregulatory processes [23].

Different researchers report that some physiological indicators, such as relative water content and photosynthetic pigment content, play an essential role in tolerance to water deficit in plants [24,25] and other adverse factors, such as extreme temperatures [26]. A reduction in chlorophyll content in the leaves could lead to harm to the photosynthetic system, consequently impacting the operation of both photosystems II and I [27]. This underscores the importance of studies on adaptive mechanisms of the photosynthetic system to gain strong insights into plant stress responses [12]. Investigations at a biochemical level of the role played by photosynthetic pigments and other variables related to physiology, such as the water status of the plant and chlorophyll content in *Lotus corniculatus*, still need to be explored. The objective of this study was to evaluate the performance of different ecotypes and a variety of *Lotus corniculatus* in terms of relative water content, concentration of photosynthetic pigments (chlorophyll *a*, chlorophyll *b*, and total chlorophyll), as well as relative chlorophyll index in response to water deficit during different seasonal times.

## 2. Results

### 2.1. Experimental Climatic Conditions

Temperature variations (T) usually affect most physiological processes of all organisms. The average temperatures recorded under shade mesh were 28.5, 23.4, 17.5, and 28.3 °C during summer (2021), autumn (2021), winter (2021–2022), and spring (2022), while the variations in relative humidity were 58.8, 47.5, 41.5 and 32.2%, respectively (Table 1). Additionally, the different ecotypes tolerated maximum temperatures of 46.9 °C and minimum temperatures of −4.6 °C, which denotes the ability to survive extreme climatic conditions.

### 2.2. Relative Water Content

The moisture content in the soil did not produce changes in the RWC among the ecotypes or the *L. corniculatus* variety that was evaluated (*p* ≤ 0.05). In general, the highest values of this variable were observed in the autumn season (Figure 1).

### 2.3. Photosynthetic Pigments

The moisture content in the soil affected the RCI among the *evaluated L. corniculatus plants*. In this study, the RCIs of 255305 and 226792 ecotypes were highest (*p* ≤ 0.05) in the autumn season with values of 244.6 and 236.5. At the same time, the Estanzuela Ganador variety registered the lowest index (147.3) under NWD conditions. The RCI did not vary in the ecotypes as a result of the moisture content in the soil during summer, winter, and spring (Figure 2).

The moisture content in the soil produced changes in *Chl a* concentration among the *L. corniculatus* ecotypes and the studied variety in the autumn, winter, and spring seasons (*p* ≤ 0.05). The 255301 ecotype recorded 187.8, 167.5, and 194.6 mg g^−1^ FW in WD, corresponding to increases of 86.0%, 172.6%, and 16.6%, respectively, in relation to the lower values obtained in the ecotype 202700 under NWD. The ecotype 226792 had the lowest response with values of 100.6, 61.4, and 81 mg g^−1^ FW in NWD. However, in the summer, no variations were observed in this pigment as a result of the effect of soil moisture content (Figure 3).

The plant genetic resources of *L. corniculatus* evaluated in this study did not show changes in chlorophyll b (*Chl b*) content (*p* ≤ 0.05) as a result of the effect of soil moisture content, which means that all ecotypes and the Estanzuela Ganador variety were not affected in terms of chlorophyll b as a result of the soil moisture content throughout the seasons (Figure 4).

The 255301 ecotype was outstanding in terms of total chlorophyll content (*p* ≤ 0.05) in WD during the autumn, winter, and spring seasons, with values of 279.4, 198.9, and 241.2 mg g^−1^ FW. In winter, the 202700 ecotype and Estanzuela variety showed high *Chl t* concentrations in NWD, with 186.9 and 192 mg g^−1^ FW. The 226792 ecotype had a lower *Chl t* concentration in NWD during the autumn, winter, and spring seasons (150.9, 72.3, and 107.3 mg g^−1^ FW), and a similar response under the same soil moisture content was observed in the 202700 ecotype during the winter (Figure 5).

The concentration of carotenoids showed significant effects (*p* ≤ 0.05) in the spring season. The highest concentration was in the 255301 ecotype under WD conditions with 53 mg g^−1^ FW. Meanwhile, the 202700 and 226792 ecotypes were lower in NWD with values of 23 and 18.7 m g^−1^ FW, respectively (Figure 6).

### 2.4. Multivariate Analysis

Principal Component Analysis (PCA) shows that the first two Principal Components (PCs) explain 71.8% of the total variance, or at least seven variables measured in the study. The first Principal Component (PC1) explained 42.1%, and the second (PC2) explained 29.7% of the variance. The structure of PC1 is integrated with the contents of *Chl a*, *Chl b*, and *Chl t* in a positive correlation; meanwhile, PC2 is made up of the variables *Car*, T, RCI, and RWC. As mentioned above, the first three variables correlate positively, and the RWC negatively connects with the group (Figure 7).

Based on Figure 8, the high cumulative percentage of the total variations observed in PC1 and PC2 is considered the evaluation factor, referring to soil water content (SWC), ecotypes, and the variety of *L. corniculatus* during the evaluation time. The 120 observations in Figure 8 are not grouped in a regular pattern according to ecotypes; however, two large groups of SWC were observed. The first group corresponds to NWD, located in the right part of Figure 8. High concentrations of *Chl a*, *Chl t*, and *Car* characterize this group. In addition, the second group is in the lower left part of the figure, and these observations correspond mainly to WD. In the groups mentioned above, the variables registered low concentrations of *Chl a*, *Chl t*, and *Car*, and healthy concentrations of RWC.

Figure 9 explains the first two PCAs, i.e., how the distribution of 120 observations of the seven variables mentioned above study correspond to the SWC during the different seasons of the year in four ecotypes and one variety of *L. corniculatus* assessed in this study. Six large groups are identified. The black circles, the upper blue triangles, the green rhombuses, and the red squares are predominantly found in the lower left part of Figure 9. These results present low levels of *Chl a*, *Chl b*, *Car*, and *Chl t*, as well as the red squares of RCI and RWC. On the other hand, three groups of observations were identified on the right side of the graph: black sum marks, inverted purple triangles, and pink triangles. A high concentration of *Car* and RCI is characterized by additional black marks. The inverted purple triangles are also remarkably high in *Chl a* and *Chl t*. Meanwhile, the pink triangles are mainly related to low *Chl b* and RWC. All these groups are associated with water deficit.

## 3. Discussion

Even though the study was carried out in shade mesh conditions, the temperature variation was extreme, with values recorded from −4.6 °C in winter to 46.9 °C in summer. This shows the survival capacity of *Lotus*’s genetic materials in extreme environments typical of the northern region of Mexico.

Developing research on the response of photosynthetic pigments in *L. corniculatus* ecotypes under water deficit is vital. Understanding this relationship provides valuable information about the possible adaptive mechanisms of plants in adverse environments, such as water-deficient conditions and extreme temperatures [28]. The genetic materials of *L. corniculatus* evaluated in this study tolerated extreme minimum and maximum temperatures. This shows the plant’s resilience to harsh environmental conditions, which indicates the potential adaptive ability of this species of *Lotus* to extreme environments [29]. The evaluation of ecotypes for tolerance to environmental stress, such as water deficit and extreme temperatures, is complex due to the integral and interaction responses in the plants [30].

Environmental temperature influences several physiological processes in plants, including pigment production and its accumulation [31]. Higher temperatures can accelerate metabolic processes, sometimes increasing pigment synthesis [32]. In this study, *L. corniculatus* showed higher concentrations of pigments at higher temperatures (Table 1) during the spring season in soil with optimal soil moisture content, although the responses to this environmental factor in plants are different depending on the plant species [31]. The 255301 and 255305 ecotypes and the Estanzuela Ganador variety presented higher concentrations of pigments when the plant was not in water-deficit conditions (Figure 8). However, excessively high temperatures can cause heat stress and negatively affect pigment production [28].

Responses at the morphological, physiological, biochemical, and genetic levels vary depending on the ecotypes used, the environment, and the ecotype–environment interactions [33]. From a physiological point of view, the flow rate of matter and energy, such as photosynthesis and transpiration, as well as stomatal conductance and relative water content (RWC), are variables frequently explored in response to water deficit [34]. The RWC is directly related to the water status of the plant [35]. In this study, the ecotypes and the studied variety of *L. corniculatus* showed a low response to RWC in both soil water content levels and showed suitable stability when the plant moved from favorable to unfavorable water conditions.

Water stress can decrease chlorophyll content [36]. This reduction in photosynthetic pigment content can be observed as yellowish or brown foliage [37]. However, in this study, the concentrations of chlorophyll *a,* chlorophyll *b,* total chlorophyll, and carotenoids in water deficit were higher than in no water deficit in most of the genetic plant materials evaluated through autumn, winter, and spring, suggesting an interesting seasonal response of adjustment. The response of an increase in pigment content was shown in the 255301 ecotype, which had a higher range of *Chl a*. Chl was shown in the 255301 ecotype, which had a higher content of *Chl a*, *Chl b*, and *Car* as a tolerance response to water deficit in all seasons except summer. These results are consistent with those reported by Mafakheri et al. [38], who found that photosynthetic pigments are highly sensitive to water deficit and low temperatures.

Other important biochemical indicators in plants’ responses to environmental stress are the concentrations of solutes and photosynthetic pigments [39]. In this study, *L. corniculatus* plants showed changes in pigment concentration in RWC with soil water content. The concentrations of these pigments decreased without stress due to the water deficit in the genetic materials evaluated. This suggests that under optimal water conditions in the soil, the tolerant activity promoted by the concentration of photosynthetic pigments is not required without water stress. However, these compounds play an essential role through the osmotic adjustment process [40] in water deficit to allow a state of turgor in the tissues in the face of the risk of plant dehydration [41].

Relative water content (RWC) is an indicator of the water status and hydration level of plants, and based on this information, the capacity to retain water in the tissues can be inferred [42]. Additionally, the influence of RWC on pigment concentration in plants can vary depending on the specific pigment, the plant species, and the adaptation strategies that the plant uses [43]. This study showed that when RWC decreased, *Car*, *Chl a*, and *Chl t* were increased. Some plants can increase their *Car* concentration as a protective mechanism against water deficit [44]. In this study, *Car* concentrations rose notably in the spring and winter seasons in the 255301 ecotype and the Estanzuela Ganador variety with no water deficit. However, a decrease in RWC can generally lead to changes in pigment concentration [42]. These changes were observed in *L. corniculatus* ecotypes; the concentration of *Chl b* decreased when RWC was low.

The RCI measures a plant’s relative concentration of chlorophyll [45,46]. During the spring, without a water deficit, the RCI is higher than in other seasons of the year. A higher RCI value indicates that higher chlorophyll content is associated with healthier and more vigorous plants [45]. RCI can monitor plant stress and track physiological changes over time.

In general, *Chl a* content increases during winter in certain plants as a physiological response to low light conditions; this phenomenon is known as winter chlorophyll increase or winter chlorophyll accumulation [47]. *L. corniculatus* ecotypes showed a high concentration of this pigment during the winter season under conditions of water deficit. The highest concentration of *Chl a* was observed in the 255301 ecotype and the Estanzuela Ganador variety under water deficit. The above suggests that the shorter days during winter and the lower angle of radiation incident on the plant result in reduced light intensity and limited exposure to sunlight. Increasing *Chl a* content in plants can maximize and improve their weak absorption capacity and photosynthetic efficiency even under low light conditions [48]. This increase in chlorophyll in winter allows perennial leaf plants such as *L. corniculatus* to continue carrying out essential metabolic processes, including photosynthesis and carbon assimilation, under limited conditions during winter [49].

On the other hand, *Chl b* is one of the primary pigments responsible for capturing light energy during photosynthesis, and can protect plants from excess light energy [50]. In this experimental trial, *L. corniculatus* ecotypes showed the highest pigment concentrations in autumn under water-deficit conditions. This could be associated with a water deficit because plants reduce water loss by closing their stomata, reducing the uptake of carbon dioxide for photosynthesis. This process can cause an imbalance between light absorption and carbon fixation, which could cause photodamage to chlorophyll molecules [51]. However, *Chl b* can help dissipate excess light energy, minimizing the potential for photodamage [50]. According to Figure 3, *Chl b* concentration decreases when T increases. These changes are part of the plant’s adaptive responses to maintaining water stress and physiological balance.

When plants are exposed to limited water availability, several changes occur in their photosynthetic pigments. Chlorophyll synthesis can be inhibited, decreasing chlorophyll content, including *Chl a* and *Chl b* [52]. This reduction in pigments can cause visible symptoms of stress (chlorosis or wilting). Despite these changes, photosynthetic pigments remain essential for capturing light energy and protecting the plant from excess light [48].

According to the multivariate analysis (Figure 1), the structure of the first two PCs showed notable bivariate associations, such that when T and RCI increase, RWC and Chl b decrease. There is an interesting case where *Car* also increased *Chl a* and *Chl t*. On the contrary, when *Car* decreased, RWC increased. Furthermore, when *Chl a* was increased, *Chl b* decreased. These results are mainly observed under conditions of optimal soil moisture content.

## 4. Materials and Methods

### 4.1. Geographic Location of the Study Area

The experiment was conducted under shade mesh conditions in the Unidad Regional Universitaria de Zonas Aridas experimental field of the Universidad Autonoma Chapingo in Bermejillo Durango, Mexico. The region is located at 25.8° NL and 103.6° WL and an altitude of 1130 m. The climate is arid, with rainfall in the summer and a cool winter, an average potential evaporation of 2000 mm, an average temperature of 21 °C, maximum of 33.7 °C and minimum of 7.5 °C, and an average annual precipitation of 258 mm [53].

### 4.2. Experimental Design

A randomized block experimental design with three repetitions was used. Ten treatments were established as a product of a 2 × 5 factorial, and thirty treatments were carried out in the experimental area. The first variation factor was the water content in the soil: no water deficit (NWD) corresponding to 100% field capacity (CC), and water deficit (WD) at 85.4% CC. The second variation factor comprised four ecotypes of *L. corniculatus* with identification codes 255301, 255305, 202700, and 226792, and the Estanzuela Ganador variety, which were procured from France, Italy, Uruguay, Canada, and Uruguay, respectively.

The experimental unit (EU) comprised one tiller of *L. corniculatus* per pot with three replicates. Previously, the seedlings were vegetatively reproduced in black plastic bags of 1 Kg capacity in soil mixed with compost. Subsequently, after two months, one plant—a rhizome with an average height of 30 cm and a tiller radius of 5 cm—was transplanted into a rigid plastic pot with 18 kg of soil capacity. The soil was a substrate prepared with a 50:30:20 soil, compost, and sand ratio. According to the physical and chemical analysis of the substrate, it had a sandy loam texture with 26% silt, 22% clay, and 52% sand, with a pH of 7.73, electrical conductivity (EC) of 7.47 dS m^−1^, and apparent density of 1.46 g cm^−3^. The experiment was carried out from March 2021 to May 2022.

The calculations of the field capacity (FC) and the permanent wilting point (PWP) were carried out using the membrane pot method using the soil moisture drawdown curve expressed in energy tension (MPa) according to Richards [54]. The field capacity (FC) was 27.5% and the permanent wilting point (PMP) was 17.5%.

### 4.3. Experimental Monitoring

The climatic conditions of temperature and relative humidity within the shade mesh were measured using a digital thermometer–hydrometer (Model OUS-WA62, ORIA, China). Irrigation was applied to each pot based on soil moisture content, which varied between 26.5% ± 1 and 23.5% ± 1 for the NWD and WD treatments. Since the field capacity was 27.5%, 23.5% of the NWD treatment average corresponded to 85.4% of FC. The soil moisture content in each pot was monitored with a real-time meter (Model MO750, Extech Instruments Co., Laredo, TX, USA).

In the first 15 days after transplanting into the pots, irrigation was at CC (27.5%) for all treatments. Subsequently, soil moisture content was restricted until the differentiated ranges of 26.5% ± 1 and 23.5% ± 1 were maintained. The irrigation intervals were defined by applying water up to the upper limit of each soil moisture treatment, corresponding to 27.5% for the WD treatment and 24.5% for the NWD treatment, and applying recovery irrigation when the lower limits for each soil moisture content were reached, corresponding to 25.5% and 22.5%, respectively.

The different genetic materials of *L. corniculatus* were homogenized. The plant canopy was cut 62 days after transplanting (DAT) with the use of pruning shears at 6 cm above the level of the potting substrate; for that, a plastic ring was used that allowed the cutting height to be standardized [55]. From that date, seasonal cutting intervals were defined: two cuts at intervals of 42 days in each seasonal period of summer, autumn, and spring, and 92 days in winter (a single cut). The latter was defined by the slow growth of the plant.

### 4.4. Measured Variables

#### 4.4.1. Relative Water Content (RWC)

The relative water content of the leaves of *L. corniculatus* was calculated using the formula proposed by Browne et al. [56], for which the saturated weight was recorded after immersing the tissue sample in distilled water for 24 h. The dry biomass was obtained by drying the plant in an oven with air circulation at 65 °C until its constant weight was reached. The results are expressed as a percentage (%) using the following formula:RWC = [((FWB − DWB))/(DWB − WSB)] × 100(1)
where RWC = relative water content, FWB = fresh weight of biomass, DWB = dry weight of biomass, and DSB = weight of saturated biomass.

#### 4.4.2. Relative Chlorophyll Index (RCI)

The measurement of this variable was carried out between the fourth and fifth leaves from the flag leaf, as suggested by De Lima Vasconcelos et al. [57], using the CM 1000 chlorophyll meter (Mod 29950, Brand Spectrum Technologies Inc., Aurora, IL 60504, USA). The measurement was performed every ten days from the cut-off between 9:00 and 10:00.

#### 4.4.3. Photosynthetic Pigments

Photosynthetic pigments were determined using the Wellburn method [58] described, in which ten fresh leaves were cut into circles of 5 mm in diameter. The leaf samples were placed in a test tube; then, 10 mL of methanol (CH_3_OH) was added and incubated in the dark at room temperature for 24 h. Absorbance (A) was measured with a spectrophotometer at 470 nm (carotenoids), 653 nm (chlorophyll b, *Chl b*), and 666 nm (chlorophyll a, *Chl a*). The calculation of the pigment concentration was carried out with the following formulas [58]:*Chl a*: [15.65·(A_666_) − 7.34·(A_653_)] [V/1000 × W](2)
*Chl b*: [27.05·(A_653_) − 11.21·(A_666_)] [V/1000 × W](3)
*Car*: [[1000 · (A_470_) − 2.86·(Chl *a*) − 129.2 (Chl *b*)]/221] [V/1000 × W](4)
*Chl t*: Chl *a* + Chl *b*(5)
where V = Extraction volume and W = fresh weight of leaf sample. The concentrations of Chl *a*, Chl *b,* and *Car* are expressed as mg g^−1^ fresh weight (FW).

### 4.5. Data Analysis

The data were analyzed with a test of means (Tukey) and Principal Component Analysis (PCA) using the statistical programs PASW Statistics for Windows, version 18.0 (SPSS Inc., Chicago, IL, USA), and Minitab Version 16.2.4 (Minitab, LLC, State College, PA, USA).

## 5. Conclusions

*L. corniculatus* ecotypes showed potential for adaptation to extreme environments of high and low temperatures throughout the seasons, with statistical differences depending on the season. The relative water content was almost stable among the evaluated ecotypes, which denotes response stability to water-deficit conditions. Total chlorophyll and carotenoid content were the leading indicators of response to water deficit in the 25301 ecotype in most seasons, except during the summer. In contrast, the 202700 ecotype and the Estanzuela Ganador variety responded during the winter, also in water deficit. The photosynthetic pigments formed a first response cluster, and the carotenoids in water deficit formed another cluster, which explains 71.8% of the variance. Some bivariate associations were identified, the first being an inversely proportional relationship between environmental temperature and the chlorophyll index with the relative content of water and chlorophyll *b.* In contrast, the content of chlorophyll *a* and total chlorophyll with carotenoid content was an inversely proportional relationship. This study provides important, crucial, essential information on the response of *Lotus corniculatus* to water deficit, which is valuable for future study programs applied to tolerance to water deficit in extreme environments.

## Figures and Tables

**Figure 1 plants-13-00961-f001:**
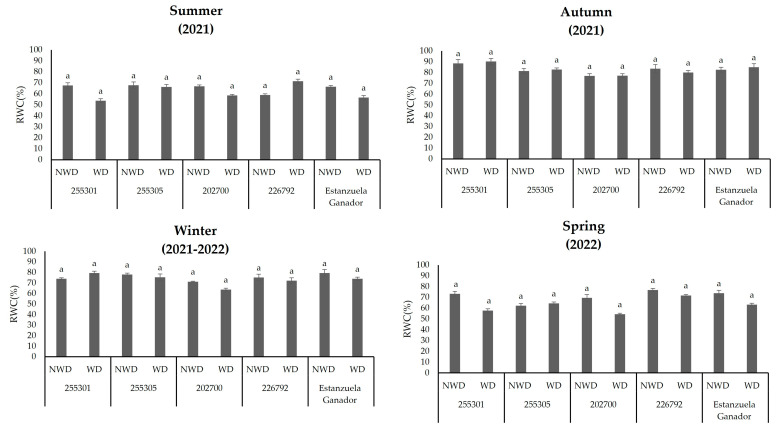
Relative water content (RWC) in different *L. corniculatus* ecotypes and the evaluated variety throughout the seasons in two soil moisture content levels. Values indicate the mean ± standard deviation. Common lowercase letters per column are not statistically significant (Tukey HSD, *p* ≤ 0.05). NWD = no water deficit; WD = water deficit. Values indicate the mean ± standard deviation. Common lowercase letters per column are not statistically significant (Tukey HSD, *p* ≤ 0.05). NWD = no water deficit; WD = water deficit.

**Figure 2 plants-13-00961-f002:**
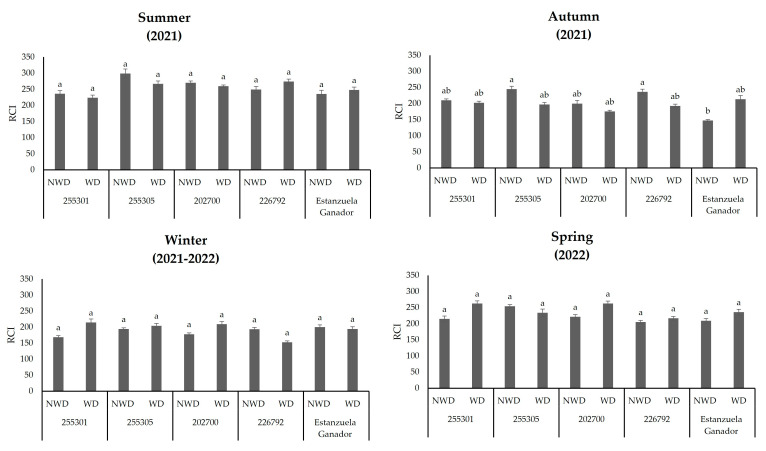
Relative chlorophyll index (RCI) in different *L. corniculatus* ecotypes and the evaluated variety throughout the seasons in two soil moisture content levels. Values indicate the mean ± standard deviation. Common lowercase letters per column are not statistically significant (Tukey HSD, *p* ≤ 0.05). NWD = no water deficit; WD = water deficit. Values indicate the mean ± standard deviation. Common lowercase letters per column are not statistically significant (Tukey HSD, *p* ≤ 0.05). NWD = no water deficit; WD = water deficit.

**Figure 3 plants-13-00961-f003:**
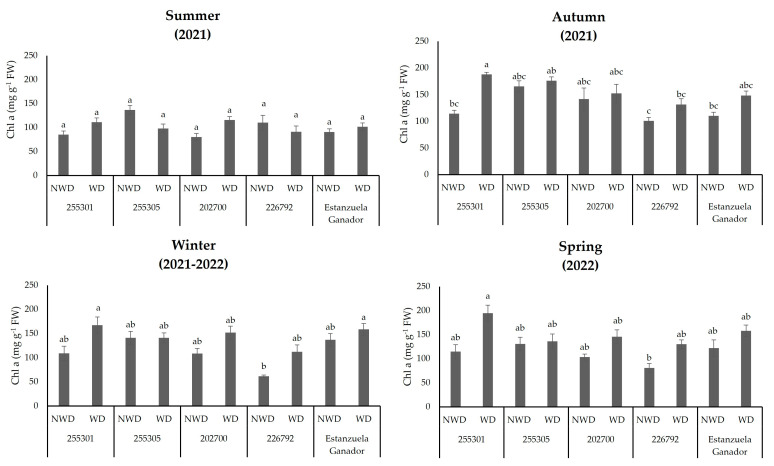
Chlorophyll a (*Chl a*) content in *L. corniculatus* ecotypes and the studied variety throughout the seasons in two soil moisture content levels. Values indicate the mean ± standard deviation. Common lowercase letters per column are not statistically significant (Tukey HSD, *p* ≤ 0.05). NWD = no water deficit; WD = water deficit.

**Figure 4 plants-13-00961-f004:**
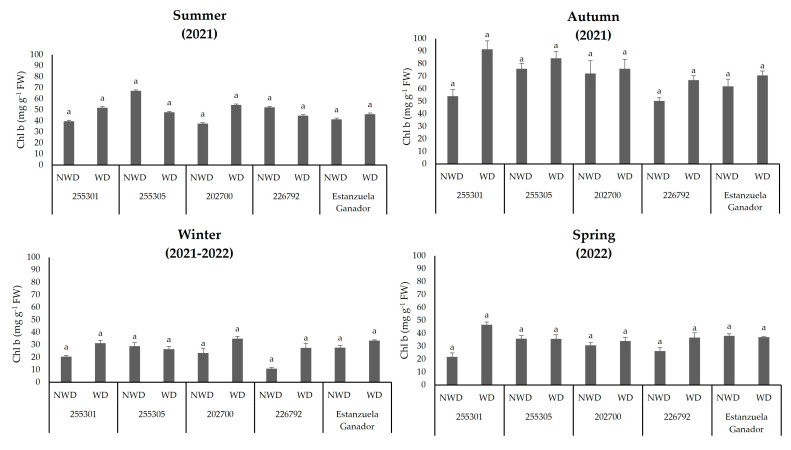
Chlorophyll b (*Chl b*) content in *L. corniculatus* ecotypes and the studied variety throughout the seasons in two soil moisture content levels. Values indicate the mean ± standard deviation. Common lowercase letters per column are not statistically significant (Tukey HSD, *p* ≤ 0.05). NWD = no water deficit; WD = water deficit.

**Figure 5 plants-13-00961-f005:**
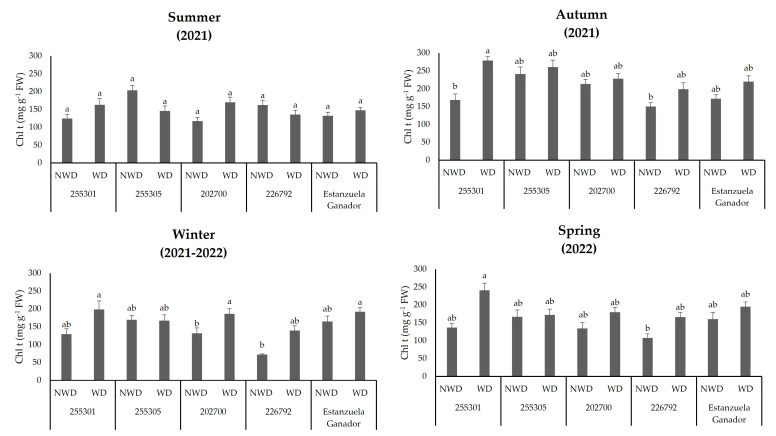
Total chlorophyll (*Chl t*) content in *L. corniculatus* ecotypes and the studied variety throughout the seasons in two soil moisture content levels. Values indicate the mean ± standard deviation. Common lowercase letters per column are not statistically significant (Tukey HSD, *p* ≤ 0.05). NWD = no water deficit; WD = water deficit.

**Figure 6 plants-13-00961-f006:**
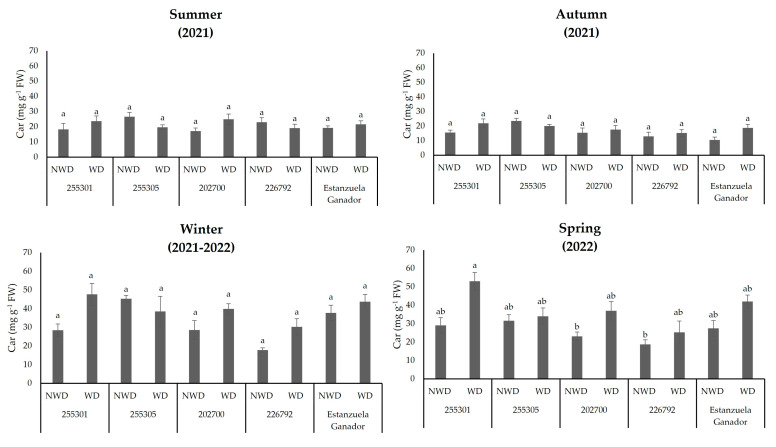
Carotenoid (*Car*) content in *L. corniculatus* ecotypes and the studied variety throughout the seasons in two soil moisture content levels. Values indicate the mean ± standard deviation. Common lowercase letters per column are not statistically significant (Tukey HSD, *p* ≤ 0.05). NWD = no water deficit; WD = water deficit.

**Figure 7 plants-13-00961-f007:**
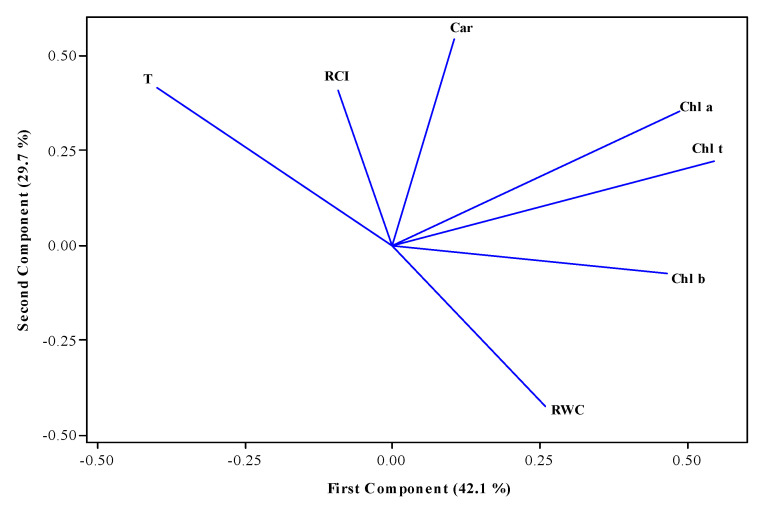
Distribution of variables—temperature (T), relative chlorophyll index (RCI), carotenoids (*Car*), chlorophyll a (*Chl a*), chlorophyll b (*Chl b*), total chlorophyll (*Chl t*), and relative water content (RWC)—in the orthogonal plane defined by the first two principal components (PCs) extracted from 120 observations as part of the experimental pots of four ecotypes and one variety of *L. corniculatus*.

**Figure 8 plants-13-00961-f008:**
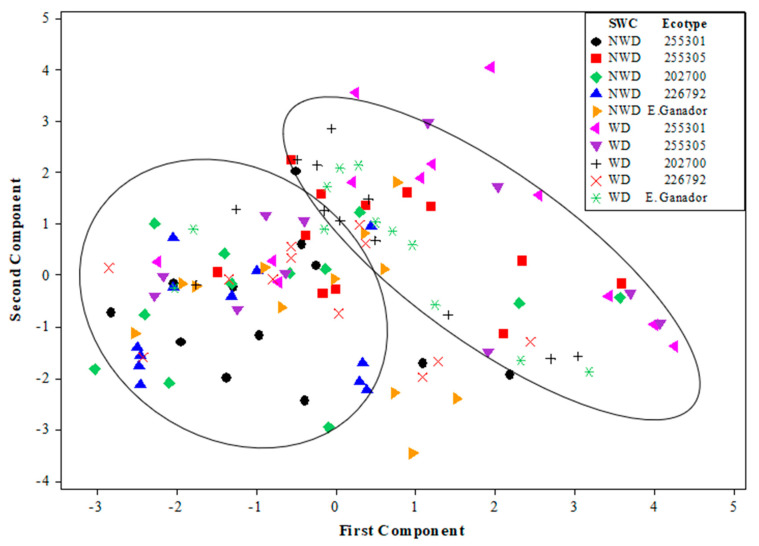
Distribution of 120 observations as part of experimental pots of four ecotypes (255301, 255305, 202700, and 226792) and one variety (Estanzuela Ganador) of *L. corniculatus* in the orthogonal plane defined by the two first principal components (PCs). The experiment was carried out over 413 days under two soil water contents (SWCs): without a water-deficit condition (NWD) of 26 ± 1.5%, and a water-deficit condition (WD) of 22 ± 1.5%.

**Figure 9 plants-13-00961-f009:**
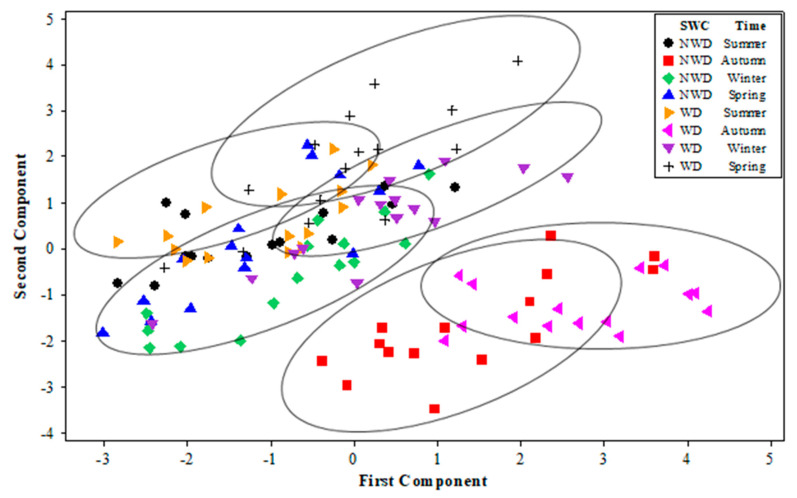
Distribution of 120 observations as part of experimental pots of four ecotypes (255301, 255305, 202700, and 226792) and one variety (Estanzuela Ganador) of *L. corniculatus* in the orthogonal plane defined by the two first principal components (PCs). The experiment was conducted over 413 days under two soil water contents (SWCs). The evaluations were completed four times, corresponding with the seasons (summer, autumn, winter, and spring).

**Table 1 plants-13-00961-t001:** The average seasonal maximum and minimum temperatures and relative humidity values inside the shade mesh during the experimental period.

Season	Temperature (°C)			Relative Humidity (%)		
	Maximum	Minimum	Mean	Maximum	Minimum	Mean
Summer (2021)	37.20	19.87	28.5	86.55	31.08	58.8
Autumn (2021)	37.02	9.85	23.4	73.29	21.83	47.5
Winter (2021–2022)	30.50	4.55	17.5	61.13	21.91	41.5
Spring (2022)	40.13	16.66	28.3	53.58	10.85	32.2

## Data Availability

Data are available on request to the corresponding author’s email with appropriate justification.

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
