# Peer review of "Relative Water Content, Chlorophyll Index, and Photosynthetic Pigments on Lotus corniculatus L. in Response to Water Deficit"

_plants, 2024, doi:10.3390/plants13070961_

Round 1

Reviewer 1 Report

Comments and Suggestions for Authors

The manuscript concerns the assessment of Lotus corniculatus ecotypes response to water deficit referring to photosynthetic pigments. Generally, changes of photosynthetic pigments to drought are well described in many studies. Therefore, it is a pity that the authors did not examine other compounds or conduct more detailed analyzes of the photosynthesis process. Other comments are listed below:

L14-32: indicate some % changes of examined parameters between control plants and plants exposed to water deficit

L35: in the Introduction briefly describe other plant responses to different abiotic stress. For example abiotic stress induced by pesticides lowered the level of carbohydrates in wheat (https://doi.org/10.1016/j.chemosphere.2022.136284). Add more similar sentences because this study is based only on photosynthetic pigments, making the study monotonous.

L114: it is a chlorophyll index, not weight

L129: indicate the correct name of the last ecotype in the Table. Why Estanzuela Ganador was not included in the Table?

L135-136: indicate some % changes between ecotypes

L335: indicate pot dimensions, how many plants per pot?

Reviewer 2 Report

Comments and Suggestions for Authors

In this manuscript (plants-2922293) entitled "Relative water content, chlorophyll index, and photosynthetic pigments on Lotus corniculatus L. as response to water deficit" submitted to Plants, Luis Angel Espíndola-González and colleagues have evaluated different L. corniculatus L. ecotypes under water deficit conditions to identify changes in the relative water content and photosynthetic pigments as indicators of physiological responses during years´s seasons. Authors suggest that soil moisture content significantly affects the performance of L. corniculatus ecotypes, and the plant shows seasonal variations in response to water deficit conditions. Collectively, this research contributes to understanding the physiological responses of L. corniculatus and its potential as a water-efficient forage crop, promoting sustainable agriculture and enhancing food security. This research is interesting and convincing, but some points need to be addressed to improve the quality of this manuscript.

1. Figure 1 appeared incorrectly twice in this manuscript. Authors should consider to show the general view of the experiment in the revised Figure 1.

2. Most of data presented in this study were collected between 2021 and 2022. What is the situation for the year 2023?

3. From the Table 1, we get informed that temperatures vary among experiment periods, which should be discussed in the revision.

4. Authors should consider to replace Tables 1-7 with Figures, which would facilitate our understanding of this study.

5. Authors need to standardize references according to the Plants template.

Round 2

Reviewer 1 Report

Comments and Suggestions for Authors

The paper was corrected. I have no more comments.

Reviewer 2 Report

Comments and Suggestions for Authors

Authors have addresses my concerns in the revision.